# Discharge Properties and Electrochemical Behaviors of Mg-Zn-*x*Sr Magnesium Anodes for Mg–Air Batteries

**DOI:** 10.3390/ma17174179

**Published:** 2024-08-23

**Authors:** Hongxuan Liu, Tingan Zhang, Jingzhong Xu

**Affiliations:** Key Laboratory of Ecological Metallurgy of Multimetal Intergrown Ores of Ministry of Education, School of Metallurgy, Northeastern University, Shenyang 110819, China; 13840520062@163.com (H.L.); xjz15140017997@163.com (J.X.)

**Keywords:** Mg-Zn-Sr alloys, Mg–air batteries, electrochemical behavior, discharge performance

## Abstract

In this work, the electrochemical and discharge properties of Mg-Zn-*x*Sr (*x* = 0, 0.2, 0.5, 1, 2, and 4 wt.%) alloys used as anodes for Mg–air batteries were systematically studied via microstructure characterization, electrochemical techniques, and Mg–air battery test methods. The addition of Sr refines the grain size, changes the composition and morphology of the passivation film and discharge products, and enhances the electrochemical properties of the alloy. Excessive Sr addition breaks the grain boundaries and precipitates a large number of Sr-rich phases, resulting in microgalvanic corrosion and the ‘chunk effect’. The anode efficiency of Mg-Zn-1Sr is the highest at a current density of 10 mA cm^−2^, reaching 61.86%, and the energy density is 2019 mW h g^−1^. Therefore, Sr is a microalloying element that can optimize the electrochemical performance of Mg–air battery alloy anodes.

## 1. Introduction

Given the frequent occurrence of global warming and El Niño phenomena, further energy conservation and emission reduction targets are urgently needed worldwide. Clean energy has gradually occupied the dominant position in the existing energy structure, and the use of new energy storage materials instead of traditional fossil fuels is a future development trend [1,2,3]. Magnesium–air batteries are considered strong competitors for the next generation of energy storage materials because of their low reduction potential (−2.37 V vs. SHE), high theoretical specific capacity (2.2 A h g^−3^), high energy density (6.8 KW h kg^−1^), low cost, good biocompatibility, and green cleanliness [4,5]. At present, the anode technology difficulty of magnesium–air batteries is that the electrons generated by anode self-discharge cannot be fully applied, and a negative differential effect (NDE) is generated, resulting in low anode utilization [6,7]. Second, the oxide film formed by the reaction covers the anode surface, reducing the anode reaction area and causing the actual discharge voltage to be lower than the theoretical value. At the same time, the hysteresis effect will be generated before the dynamic balance of oxide film formation and breakage [8,9]. Third, the occurrence of self-corrosion and the rupture of the oxide film will cause some of the unreacted anodes to fall off, that is, the ‘chunk effect’ [9,10].

Alloying is a common means to improve the electrochemical performance of magnesium–air battery anodes. Common binary magnesium alloy systems include the Mg-Al system, Mg-Zn system, and Mg-Li system. Common ternary magnesium alloy systems include the AZ system, AP system, and WE system. Four-component and five-component magnesium alloys have also been widely studied by researchers. The influence of different microalloying elements on the matrix and the synergistic competition mechanism between different elements are the main directions of current research [11,12]. Mg-Zn alloys have the advantages of good stability, strong corrosion resistance, high battery efficiency [13,14,15], and a unique long-period ordered stacking phase (LPSO). They have been extensively studied in the field of anode microalloying in magnesium–air batteries.

Sr is less electronegative than Mg is, and electrons are easier to remove. A more negative potential with the second phase generated by Mg can effectively improve the discharge performance of magnesium alloys. The addition of Sr can also refine the grains and improve the corrosion resistance and anode utilization of the alloy. During the discharge process, more Sr can form loose and easy-to-fall surface corrosion products, have a larger reaction contact area, and reduce the occurrence of pitting at the grain boundaries, which slows the corrosion rate of the alloy [12,16]. Xiang [17] studied the synergistic effect of Sr on the corrosion resistance and refinement of Mg alloys via a galvanic corrosion model. Sr replaced the byproduct of hydroxyapatite and improved the corrosion resistance of the alloy. Chen [18] reported that adding Sr to a Mg–Zn alloy can achieve grain refinement, reduce the fraction of intermetallic compounds at grain boundaries, and increase corrosion resistance. Nayeri [19] reported that the addition of Sr preferentially formed a second phase of Sr, which affected the composition and proportion of the original intergranular compounds and had a great influence on the corrosion resistance. Yu [20] studied the effect of Sr on the Mg matrix. Mg-1Sr showed the largest open circuit potential of −1.6961 V, the highest anode efficiency of Mg-0.2Sr reached 60.18%, and the specific capacity was 1342.28 mA h g^−1^.

The effect of Sr on the corrosion resistance of Mg-Zn alloys has been extensively studied in the field of biomedicine. However, there are few reports in the field of battery-related materials, and the effect of Sr on the electrochemical properties of Mg-Zn alloys is still unclear. In this work, six different compositions of Mg-5Zn-*x*Sr alloys (0–4 wt.%, hereinafter referred to as Mg-Zn-*x*Sr) were prepared. The effects of Sr on the microstructure, electrochemical behavior, and discharge performance of the Mg-5Zn alloy were studied and compared with those of the Mg-5Zn alloy. The influence of the second phase containing Sr is revealed by the microstructure, which provides the data basis for the microalloying design of magnesium alloy anodes.

## 2. Materials and Methods

### 2.1. Materials Preparation

In this study, Mg-Zn-*x*Sr (*x* = 0, 0.2, 0.5, 1, 2, and 4) alloys were prepared by casting pure magnesium (99.9 wt.%), pure zinc (99.9 wt.%), and Mg-30Sr master alloys. Figure 1 is the flow chart of the experiment, the well-proportioned raw materials were placed in a high-purity graphite crucible and continuously heated to 750 °C in a relative vacuum shaft furnace, during which Ar and SF_6_ protective gasses were continuously introduced. The molten metal was homogenized at 720 °C for 30 min and poured into a low-carbon steel mold with a diameter of 25 mm and a height of 200 mm. The mold was preheated at 300 °C for 3 h in advance. Table 1 lists the actual composition of the alloy analyzed via inductively coupled plasma (ICP) emission spectrometer (Prodigy Plus, Salt Lake City, UT, USA) (pump rate: 60 r/min; plasma gas: 12.0 L/min; nebulizer flow: 0.70 L/min; stable time: 20 s; auxiliary gas: 1.0 L/min; reading access time: 5 s; sample flush time: 20 s; RF power: 1250 W).

### 2.2. Microscopic Characterization

The microstructure of the Mg-Zn-xSr alloy was characterized via optical microscopy (OM, Laica DM4P, Wetzlar, Germany) and scanning electron microscopy (SEM, Zeiss Sigma 300, Oberkochen, Germany) with an energy dispersive spectrometer (EDS). Before observation, the samples were ground with SiC sandpaper with particle sizes of 600, 1000, 2000, and 3000 and polished with Cr_2_O_3_ polishing paste to remove most of the scratches. The surface of each sample was chemically etched with 10% HNO_3_, ultrasonically cleaned in ethanol, and air-dried. The processed samples were photographed to obtain OM images. The grain size of the Mg-Zn-xSr alloy was measured via the ImageJ Pro Plus linear intercept method. Each alloy was analyzed in six different areas to measure the average grain size. The phase composition of the alloy was identified via X-ray diffraction (XRD) (CuKα radiation, range of 10–90° (2θ), 0.02° scanning step, Bruker D8 Advance, Karlsruhe, Germany).

### 2.3. Electrochemical Measurement

A traditional three-electrode system was used to test the electrochemical performance with an electrochemical workstation (Zahner Zennium, ZAHNER, Kronach, Germany). The reference electrode was a saturated Ag/AgCl electrode, and the inert electrode was a Pt electrode. The prepared Mg-Zn-*x*Sr alloy was placed in an epoxy resin conductive mold with an exposed area of 10 mm × 10 mm. As a working electrode, it is also an anode in the system. Before placement, the samples were polished according to the same steps as those used for microstructure observation. The electrolyte used in the electrochemical test was a 3.5 wt.% NaCl solution. Before the test, the sample was put into the solution for 30 min to obtain a stable open circuit potential (OCP). The potential dynamic polarization test range was from −400 mV to 400 mV (vs. OCP), and the scanning rate was 0.5 mV s^−1^. The corrosion current density was calculated via extrapolation of the corrosion potential from the polarization curve. Electrochemical impedance spectroscopy (EIS) was also performed using the same equipment. The scanning frequency range was 100 kHz to 0.01 Hz, and the voltage amplitude was 5 mV. ZSimpWin3.60 software was used to fit the EIS data. To ensure the repeatability of the experiment, each anode material was subjected to three repeated experiments.

### 2.4. Mg–Air Battery Measurement

In the NEWARE battery test system, the discharge performance of the studied alloy was analyzed via a commercial Mg–air battery (full battery) module. A commercial air cathode (Suzhou, China) composed of a MnO_2_/C catalyst and a nickel (Ni) conductive network was used as the cathode to test the discharge curve of the Mg–air battery under a constant current density in a 3.5 wt.% NaCl solution. The applied current densities were 0.5 mA cm^−2^, 2 mA cm^−2^, 5 mA cm^−2^, and 10 mA cm^−2^, and the discharge experiment was performed for 10 h. The surface morphology of the material after discharge was characterized via SEM. After the surface product was removed with 200 g/L chromic acid, the anode utilization, specific capacity and energy density of the Mg-Zn-*x*Sr alloy were calculated via Equations (1), (2) and (4). All tests were performed at least three times to ensure repeatability.
(1)$$Utilization\;Efficency\left(\%\right)=\frac{W_{theo}}{∆W}\times 100\%$$
(2)$$Specific\;capacity\left(mA\;h\;g^{-1}\right)=\frac{I\times t}{∆W}\times 1000$$
where *I* is the discharge current (*A*), *t* is the discharge time (h), and ∆W(g) is the weight loss of the sample during the reaction, which can be measured as the mass of the sample before and after the reaction. $W_{theo}\;$(g) shows the theoretical weight loss calculated on the basis of Faraday’s theory. It can be calculated via Formula (3):(3)$$W_{theo}=\frac{I\times t}{F\times \sum \left(\frac{x_{i}\times n_{i}}{m_{i}}\right)}$$
where *F* is the Faraday constant (26.8 A h mol^−1^), and *x_i_*, *n_i_*, and *m_i_* are the mass fraction of each element in the material, the number of exchanged electrons, and the atomic weight, respectively.
(4)$$Specific\;energy\left(W\;h\;{kg}^{-1}\right)=\frac{\int_{0}^{t}U\times I\times ∆t}{∆W}$$
where *U* is the discharge voltage (V), *I* is the discharge current (A), *t* is the discharge time (h), and ∆W is the weight (kg) lost by the anode during the discharge process.

## 3. Results and Discussion

### 3.1. Microstructures of the Mg-Zn-xSr Alloys

Figure 2 shows the XRD patterns of the Mg-Zn-*x*Sr alloys (*x* = 0, 0.2, 0.5, 1, 2, and 4 wt.%).

The main phase composition of the alloy is the α–Mg phase, which is a solid solution diffusion phase formed during diffusion when the concentration of Zn and Sr on the surface of the magnesium alloy has not yet reached the required concentration of the compound. When Sr is not added, the phase is mainly composed of α-Mg, MgZn, and Mg_7_Zn_3_. The Mg_17_Sr_2_ phase appears when 0.2 wt.% Sr is added, and the peak value gradually increases with the addition of Sr. The Mg_11_Zn_4_Sr_3_ phase appeared when the addition amount reached 1 wt.%, and the peak phase was the most obvious when the addition amount reached 4 wt.%. This is consistent with the Mg-Zn-Sr ternary phase diagram [21].

The addition of Sr has a significant effect on the grain size, morphology, and second volume fraction. Figure 3 shows the OM image of the Mg-Zn-*x*Sr alloy after corrosion.

Table 2 lists the average size and refinement rate of the alloy.

The solid solubility of Sr in Mg is 0.11 wt.%. When the solid solubility limit is exceeded, the second phase between the metals separates to the grain boundaries, and grain refinement is induced by the grain growth constraint mechanism. The addition of Sr increases the degree of undercooling at the front of the solid–liquid interface, which can increase the number of nuclei in the primary α-Mg phase [18,22]. As a surface-active element, Sr is enriched in front of the solid/liquid interface during the solidification process of the alloy, which hinders grain growth and refines the divorced eutectic structure [23]. The average grain size of the Mg-Zn alloy is 176 μm, and the second phases, MgZn and Mg_7_Zn_3_, are dispersed in the grains. With the addition of 0.2 wt.% Sr, the grain size decreases to 146 μm, and the refinement ratio is 17.04%. With the addition of Sr, the average size of the alloy decreases gradually, but the refinement effect decreases gradually. At the same time, the MgZn and Mg_7_Zn_3_ phases gradually distribute from the point to the grain boundaries. When 1 wt.% is added, the grain size is the most stable and uniform, and many Sr second phases distributed along the grain boundaries appear in the alloy, which inhibits the recovery of dislocations and weakens the effect of grain refinement [24]. When the addition amount of Sr reaches 4 wt.%, the average grain size is 70 μm, but the size deviation reaches 77.14%, and the grain boundaries become rough and irregular. This may be because the trace of the second phase is completely redissolved during the homogenization process, which accelerates the migration of the grain boundaries. Many second phases precipitate from the grain boundaries, resulting in obvious dendrites [20]. Moreover, structural undercooling occurs [25], many nucleation sites are activated, and the size distribution is very uneven, which easily causes microgalvanic corrosion and reduces the corrosion resistance of the alloy. The corroded alloy was characterized via SEM and EDS. The results are shown in Figure 4.

The distribution of the grain size is basically consistent with the results shown in the OM images. There is a small amount of precipitated second phase in the Mg–Zn alloy, and the composition is mainly Mg_7_Zn_3_, which corresponds to the black dots observed in the grain interior in the OM image. In the Mg-Zn-0.2Sr alloy, the content of Sr is low and still exists as a dot-like second phase, most of which is Mg_17_Sr_2_, and there is a small amount of Mg_11_Zn_4_Sr_3_; at this time, the grain size distribution is relatively uneven. Many Mg_11_Zn_4_Sr_3_ phases began to appear in the Mg-Zn-0.5Sr sample; the grain size tended to be uniform, and the punctate second phase began to diffuse gradually to the grain boundaries. Obvious grain boundaries were produced in Mg-Zn-1Sr, and the Mg_11_Zn_4_Sr_3_ phase completely diffused to the grain boundaries, forming a network structure. The grain size gradually decreases, and the distribution is very uniform, which can effectively hinder the diffusion of corrosion [26]. When the Sr content is 2 wt.%, many Mg_11_Zn_4_Sr_3_ phases precipitate, which are distributed in the interior of the grains and around the grain boundaries. The grain boundaries begin to break and become irregular [27]. Moreover, the grain size is slightly enlarged, and the size distribution is uneven, which corresponds to the OM image. When 4 wt.% Sr is added, the grain boundaries are almost completely broken, forming a nanoscale Sr-rich second phase in an agglomerated state with a high Sr content. The grain size is very uneven, and there are multiple pitting sites, which has a great influence on corrosion resistance.

### 3.2. Electrochemical Analysis of Mg-Zn-xSr Alloys

The open-circuit potential curve of the Mg-Zn-*x*Sr alloy measured in a 3.5 wt.% NaCl solution for 7200 s is shown in Figure 5a.

After immersion in the solution, the open-circuit potential of the alloy changes rapidly. With increasing immersion time, all anode potentials tend to be stable at 200–600 s, which is attributed to the anodic polarization caused by the formation of passive films and corrosion products on the surface [28]. The average OCP after stability is set to E_oc,_ as listed in Table 3. The time for strontium-containing samples to reach stability is shorter, and the negative potential is greater, which is attributed to the fact that Sr has a more negative standard electrode potential than do Mg and Zn. Dissolution into the α-Mg matrix can reduce the electrode potential, refine the grains, and increase the electrochemical activity. When the addition amount reaches 1 wt.%, the negative potential is the largest, reaching −1.689 V (vs. Ag/AgCl), and the continuous addition of negative potential decreases sharply, which may be due to the different rates of the cathodic hydrogen evolution reaction.

Figure 5b shows the potentiodynamic polarization curves of the Mg-Zn-*x*Sr alloys. The anodic polarization curve is related to the dissolution of Mg, whereas the cathodic polarization curve is attributed to the hydrogen evolution reaction. Strong passivation can be observed in the anode branch, which means that the anode film forms and acts as a physical shielding layer to prevent self-discharge of the anode [29,30]. With the addition of Sr, the cathodic polarization curve continues to move toward a high current, and the cathodic reaction kinetics are accelerated. Table 3 lists the electrochemical parameters during the reaction of the Mg-Zn-*x*Sr alloy.

E_oc_ is calculated from the average open circuit potential after stabilization. E_corr_ and I_corr_ are obtained from the Tafel extrapolation fitting results. The self-corrosion potential indicates the thermodynamic tendency of the reaction, that is, the electrochemical activity. The Mg-Zn-1Sr alloy has the largest E_corr_, indicating that the electrochemical activity is the strongest because the increase in the number of grain boundaries promotes dissolution of the Mg matrix, which also corresponds to the largest E_oc_. The self-corrosion current density indicates that the kinetics of the reaction are also related to the corrosion rate. The larger the I_corr_ is, the faster the corrosion rate and the lower the corrosion resistance. The corrosion current of the Mg-Zn-4Sr alloy reaches 102.67 μA cm^−2^, and the corrosion rate is the highest, which may be related to the microgalvanic corrosion caused by the uneven grain size.

EIS is mainly used to analyze the interface reaction process between the electrode and the electrolyte and the surface condition of the electrode. The results are shown in Figure 6.

The Nyquist diagram shows that the Mg-Zn-0.5Sr, Mg-Zn-1Sr, and Mg-Zn-2Sr alloys exhibit two large capacitance loops of high frequency and medium frequency over the whole frequency range. The high-frequency capacitance loop is generated mainly by the oxide film on the surface of the alloy. This film inhibits the mass diffusion process, resulting in a large diffusion overpotential. The high-frequency loop exhibits a contraction state at high Sr contents, indicating that the surface oxide film has poor protection [31]. The intermediate frequency loop is used to study the charge transfer process. The small intermediate frequency loop indicates that the resistance is low and that the charge transfer overpotential decreases, whereas the low frequency induction loop is caused mainly by the dissolution and desorption of the surface film [32]. The Mg-Zn-1Sr alloy has the largest charge transfer overpotential and diffusion overpotential, so it has the highest discharge potential and the largest OCP. Figure 6b shows the fitting data of different components in the equivalent circuit in ZSimpWin software. *R_s_* represents the solution resistance, and the film resistance (R*_f_*) and the film capacitance (CPE*_f_*) are used to present the first capacitance loop in the equivalent circuit. The intermediate frequency loop is determined by the charge transfer resistance (R*_ct_*) and the electric double layer capacitance (CPE*_dl_*) [33]. The Mg-Zn-1Sr alloy has a relatively high R*_f_* and R*_ct_*, indicating that the original surface oxide film is dense and has few voids, which is beneficial for preventing anode corrosion. In the high-Sr alloy, the increase in the second phase accelerated the hydrogen evolution reaction, promoted cracking of the corrosion products, and reduced the R*_f_* and R*_ct_*.

### 3.3. Discharge Properties of the Mg-Zn-xSr Alloys

The prepared Mg-Zn-*x*Sr alloys were used as the anode in the Mg–air battery for the discharge test to evaluate the discharge potential of different alloy anodes. The constant current discharge curves at current densities of 0.5, 1, 5, and 10 mA cm^−2^ were obtained in 3.5 wt.% NaCl solution, as shown in Figure 7.

When the anode is just placed in the electrolyte, the contact area between the surface and the electrolyte is large, resulting in a large discharge voltage. As the discharge progresses, a Mg(OH)_2_ film is formed on the anode surface, which hinders contact between the surface and the electrolyte and hinders ion transport, and the discharge voltage decreases [34,35]. As the discharge process progresses, the formation and rupture of the surface oxide film reach dynamic equilibrium, and the voltage tends to be stable, which is also the reason for the hysteresis effect. At 10 mA cm^−2^, the voltage fluctuates violently, which is caused by the balance between the deposition and separation of the discharge products [36]. The addition of Sr to the Mg-Zn alloy can increase the discharge voltage, but when the addition amount reaches 4 wt.%, the voltage decreases sharply, and the discharge process is extremely unstable, which may be related to the microgalvanic corrosion caused by the small size of the alloy. Among all the alloys, Mg-Zn-1Sr has the best discharge potential, and the discharge process is stable. At a current density of 0.5 mA cm^−2^, the discharge voltage reached −1.607 V. At a current density of 10 mA cm^−2^, the anode utilization rate reached 61.86%.

Moreover, the anode utilization rate and specific capacity of the alloy during discharge are calculated via Equations (1), (2) and (3). The calculation results are listed in Table 4.

A small amount of added Sr can significantly improve anode utilization, specific capacity, and energy density because a denser and more protective Sr-containing oxide layer will be formed. With galvanic corrosion between the second phase and the matrix, the protective film fails [37,38]. The discharge voltages of all the anodes decreased significantly at a high current density (10 mA cm^−2^). This occurred because an increase in the current density increases the rate of the self-corrosion reaction and accelerates the hydrogen evolution reaction, but the anode efficiency tends to increase. The Mg-Zn-0.5Sr alloy exhibited a maximum discharge voltage of −1.608 V at 0.5 mA cm^−2^. The Mg-Zn-1Sr alloy exhibited the best anode performance among all the anodes, with the highest energy density of 2046 mW h g^−1^ at 5 mA cm^−2^, which proves that it has the best corrosion resistance. This may be related to the uniform grain size distribution and the formation of network grain boundaries to prevent rapid corrosion diffusion. The anode efficiency, specific capacity, and energy density are also related to the ‘chunk effect’ during the discharge process. Owing to uneven corrosion, some unreacted anode metal particles fall off directly from the surface and cannot participate in the subsequent discharge process, which greatly affects the utilization efficiency, specific capacity, and energy density of the anode.

### 3.4. Surface Morphologies after Discharge

The surface morphology after anode discharge and removal of discharge products on the anode surface can provide more details of anode dissolution during discharge, which is conducive to elucidating the activation mechanism of the anode. An SEM image of the anode after discharge at a current density of 2 mA cm^−2^ for 2 h is shown in Figure 8. The surface of the alloy with a low Sr content is covered by dense discharge products, which can prevent the penetration of the solution and reactants, whereas the discharge products on the anode surface of the alloy with a high Sr content are looser and more porous. There are many protrusions in the discharge products on the anode surface, which are caused by the local dissolution of the Mg matrix. Among them, there are many discharge product residues and protrusions on the surface of the Mg-Zn and Mg-Zn-4Sr anodes, while the surface of the Mg-Zn-1Sr anode is flattest. Uniform dissolution effectively prevents the occurrence of the ‘chunk effect’, and the discharge products are nanophases, indicating that the discharge reaction is intense and that the discharge products are peeled off quickly. The presence of holes on the surface of Mg-Zn-0.2Sr was observed, which may be caused by the direct shedding of unreacted particles. The unreacted second phase of Sr still existed on the surface of the Mg-Zn-2Sr anode, and the Mg matrix was preferentially dissolved. The presence of many second phases results in uneven corrosion and causes localized corrosion. There are many deep pits on the surface of the Mg-Zn-4Sr anode, which may be caused by the chunk effect, and the distribution of reaction products is not uniform, so there may be microgalvanic corrosion.

Figure 9 shows the SEM image after the discharge product was removed.

The anode surface presents different morphologies. Mg-Zn, Mg-Zn-0.5Sr, Mg-Zn-1Sr, and Mg-Zn-2Sr dissolve more evenly, which easily peels off the discharge product. The small grooves on the surface are related to the release of hydrogen. The surfaces of Mg-Zn-0.2Sr and Mg-Zn-4Sr are rough, and many undissolved substrates and pores exist. Mg-Zn-4Sr has a layered dissolution morphology. The complex structure increases the difficulty of shedding discharge products, and many precipitated second phases accumulate at the grain boundaries, which aggravates uneven dissolution and leads to the ‘bulk effect’, and the anode utilization rate is greatly reduced. There are second-phase grain boundaries similar to ribbons in Mg-Zn-1Sr and Mg-Zn-2Sr, which can prevent rapid corrosion and produce many cracks and broken areas, which help the electrolyte penetrate the matrix and activate the discharge reaction.

In order to further explore the application potential of Mg-Zn-xSr alloy as a Mg–air battery, we compared the Mg-Zn-1Sr alloy with the best discharge performance with commercial AZ31 alloy [39] and commercial AM10 [32]. The results are shown in Table 5. At a current density of 5 mA cm^−2^, the Mg-Zn-1Sr alloy exhibits excellent discharge performance. Compared with AZ31 and AM10, the discharge voltage is increased by 15.5% and 11.7%, the anode utilization rate is increased by 42.7% and 36.4%, and the energy density is increased by 25.7% and 14%, which is a favorable competitor for anode materials of Mg–air batteries.

## 4. Conclusions

In this work, the microstructure and electrochemical and discharge properties of Mg-Zn-*x*Sr anodes in Mg–air batteries were systematically studied. The main results are as follows:The addition of Sr refined the grain size and improved the discharge performance of the Mg–Zn alloy. With the addition of Sr, the original point-like second phase gradually moved to the grain boundaries, and obvious grain boundaries formed when the Sr content reached 1 wt.%. As the Sr content continues to increase, the grain boundaries tend to crush, a large number of Sr-rich second phases begin to precipitate, the grain size becomes uneven, and the pitting points increase, resulting in the occurrence of galvanic corrosion.The addition of Sr can enhance the corrosion resistance of the matrix, which is due to the formation of a denser protective film and a network of grain boundary structures, hindering grain growth so that the grain size tends to be uniform. In alloys with high Sr contents, anodic dissolution is very uneven, resulting in many complex deep pits and layered protrusions, which is not conducive to the shedding of corrosion products and produces a ‘bulk effect’, which has a great influence on the electrochemical performance.The Mg-Zn-1Sr alloy exhibited the best discharge performance, with an open circuit potential of −1.689 V. In the Mg–air battery, the maximum energy density was 2046 mW h g^−1^ at a current density of 5 mA cm^−2^, and the maximum anode utilization rate was 61.86% at a current density of 10 mA cm^−2^, which is an excellent alternative material for Mg–air battery anode materials.

## Figures and Tables

**Figure 1 materials-17-04179-f001:**
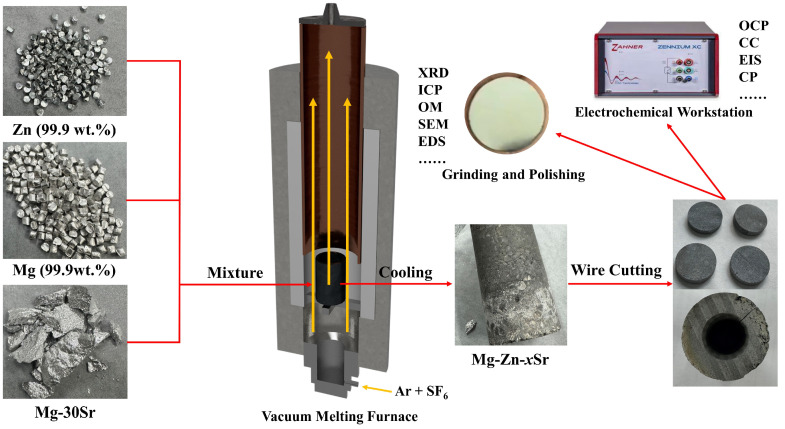
Experimental flow chart.

**Figure 2 materials-17-04179-f002:**
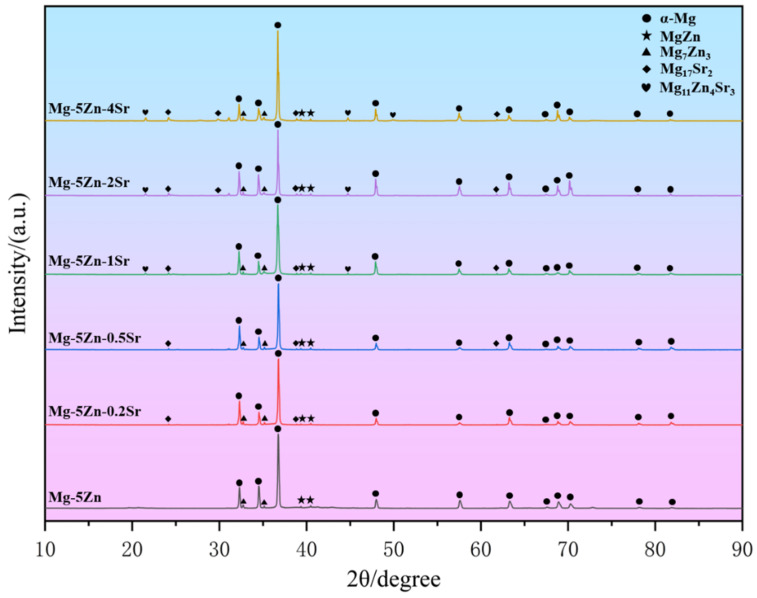
XRD patterns of the Mg-Zn-*x*Sr alloys (*x* = 0, 0.2, 0.5, 1, 2, and 4 wt.%).

**Figure 3 materials-17-04179-f003:**
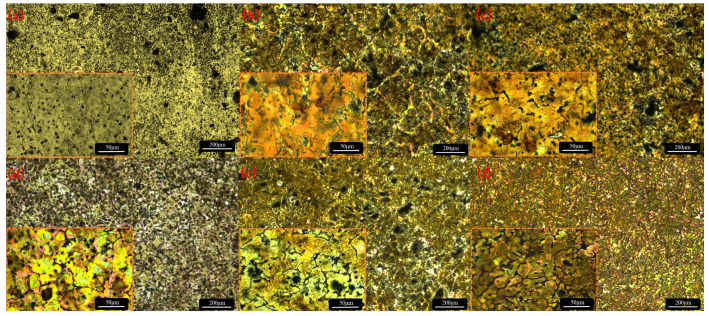
OM images: (**a**) Mg-5Zn; (**b**) Mg-Zn-0.2Sr; (**c**) Mg-Zn-0.5Sr; (**d**) Mg-Zn-1Sr; (**e**) Mg-Zn-2Sr; (**f**) Mg-Zn-4Sr.

**Figure 4 materials-17-04179-f004:**
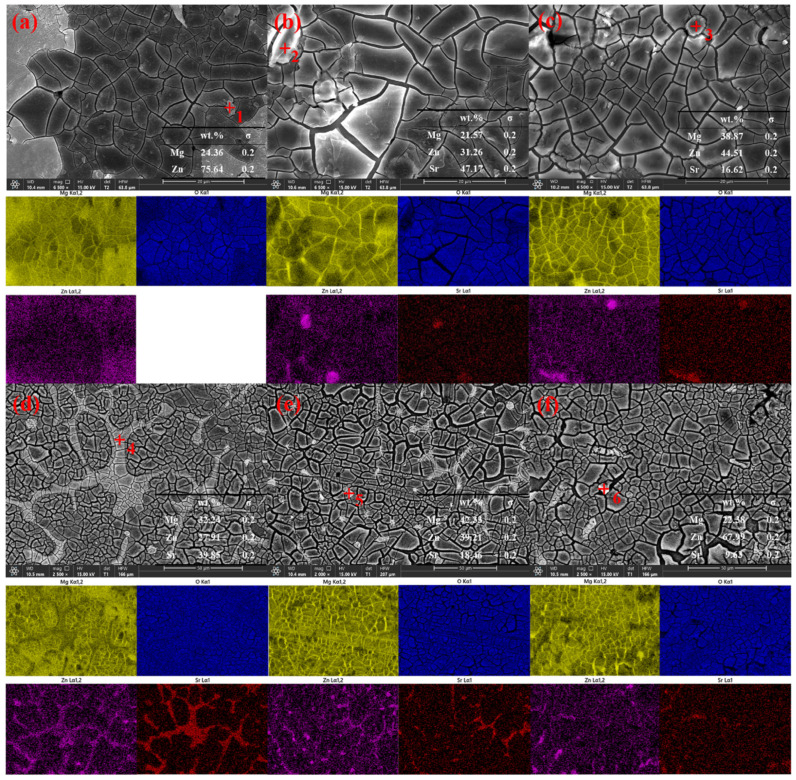
SEM micrographs (**a**) Mg-Zn; (**b**) Mg-Zn-0.2Sr; (**c**) Mg-Zn-0.5Sr; (**d**) Mg-Zn-1Sr; (**e**) Mg-Zn-2Sr; (**f**) Mg-Zn-4Sr. Numbers represent the distribution of EDS results of different elements in SEM results.

**Figure 5 materials-17-04179-f005:**
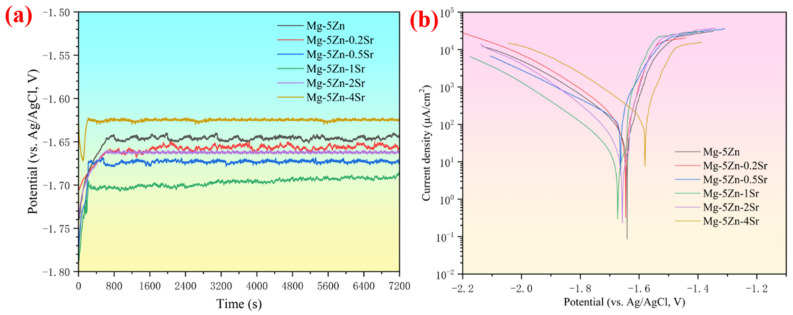
Electrochemical properties of the Mg-5Zn-*x*Sr alloy in 3.5 wt.% NaCl: (**a**) OCP and (**b**) polarization curve.

**Figure 6 materials-17-04179-f006:**
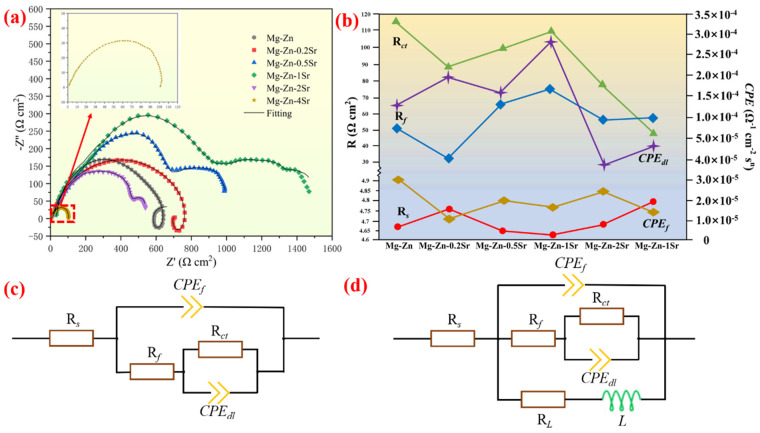
(**a**) Nernst chart; (**b**) fitting of the original parameters; (**c**) EIS equivalent circuits of the Mg-Zn, Mg-Zn-0.2Sr, and Mg-Zn-4Sr alloys; (**d**) EIS equivalent circuits of the Mg-Zn-0.5Sr, Mg-Zn-1Sr, and Mg-Zn-2Sr alloys.

**Figure 7 materials-17-04179-f007:**
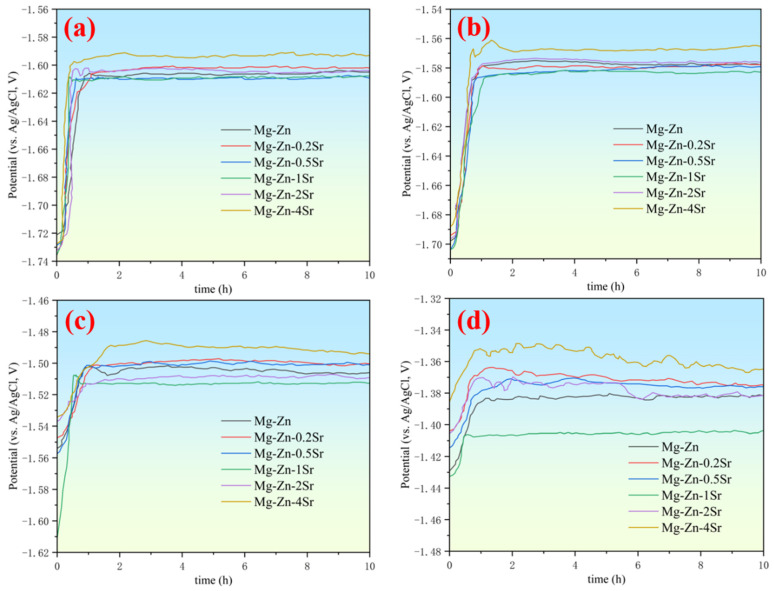
Discharge curves of the investigated anodes in 3.5 wt.% NaCl solution at current densities of (**a**) 0.5 mA cm^−2^, (**b**) 2 mA cm^−2^, (**c**) 5 mA cm^−2^, and (**d**) 10 mA cm^−2^.

**Figure 8 materials-17-04179-f008:**
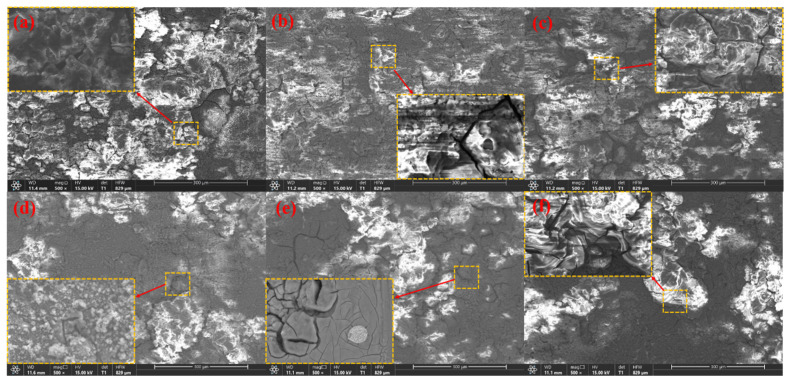
Micromorphology of surface products after discharge for 2 h at 2 mA cm^−2^: (**a**) Mg-Zn; (**b**) Mg-Zn-0.2Sr; (**c**) Mg-Zn-0.5Sr; (**d**) Mg-Zn-1Sr; (**e**) Mg-Zn-2Sr; (**f**) Mg-Zn-4Sr.

**Figure 9 materials-17-04179-f009:**
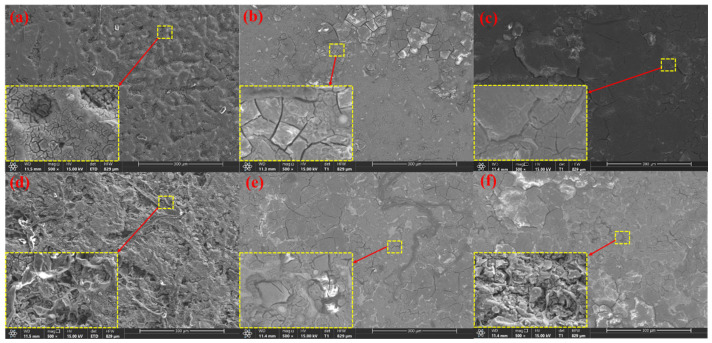
Micromorphology of the surface products removed after 2 h of discharge at 2 mA cm^−2^: (**a**) Mg-Zn; (**b**) Mg-Zn-0.2Sr; (**c**) Mg-Zn-0.5Sr; (**d**) Mg-Zn-1Sr; (**e**) Mg-Zn-2Sr; (**f**) Mg-Zn-4Sr.

**Table 1 materials-17-04179-t001:** Chemical composition of the Mg-Zn-*x*Sr alloys (wt.%).

Materials	Zn	Sr	Fe	Al	Mn	Si	Ni	Cu	Mg
Mg-5Zn	4.982	-	0.036	0.014	0.0124	0.0011	0.0002	0.0012	Bal.
Mg-5Zn-0.2Sr	4.836	0.195	0.012	<0.01	0.0083	0.0121	0.0007	0.0016	Bal.
Mg-5Zn-0.5Sr	5.064	0.524	0.016	<0.01	0.0034	0.0027	0.0006	0.0009	Bal.
Mg-5Zn-1Sr	4.921	1.064	0.021	0.016	0.0056	0.0020	0.0006	0.0021	Bal.
Mg-5Zn-2Sr	5.027	1.967	0.024	0.008	0.0021	0.0041	0.0009	0.0013	Bal.
Mg-5Zn-4Sr	5.133	4.126	0.007	0.021	0.0049	0.0016	0.0004	0.0008	Bal.

**Table 2 materials-17-04179-t002:** Average grain size of the Mg-Zn-*x*Sr alloys.

Materials	Mg-Zn	Mg-Zn-0.2Sr	Mg-Zn-0.5Sr	Mg-Zn-1Sr	Mg-Zn-2Sr	Mg-Zn-4Sr
Grain size (μm)	176 ± 87	146 ± 88	97 ± 37	77 ± 26	68 ± 34	70 ± 54
Refinement ratio	0	17.04%	44.88%	56.25%	61.36%	60.23%

**Table 3 materials-17-04179-t003:** Electrochemical parameters of the Mg-5Zn-*x*Sr alloys.

Material	E_oc_ (V vs. Ag/AgCl)	E_corr_ (V vs. Ag/AgCl)	I_corr_ (μA cm^−2^)
Mg-5Zn	−1.646	−1.644	29.53
Mg-5Zn-0.2Sr	−1.659	−1.648	41.12
Mg-5Zn-0.5Sr	−1.672	−1.662	91.52
Mg-5Zn-1Sr	−1.689	−1.669	16.48
Mg-5Zn-2Sr	−1.664	−1.651	32.75
Mg-5Zn-4Sr	−1.625	−1.579	102.67

**Table 4 materials-17-04179-t004:** Discharge properties of the Mg-Zn-*x*Sr alloys.

	Current (mA cm^−2^)	Mg-Zn	Mg-Zn-0.2Sr	Mg-Zn-0.5Sr	Mg-Zn-1Sr	Mg-Zn-2Sr	Mg-Zn-4Sr
Discharge potential (vs. Ag/AgCl, V)	0.5	−1.605	−1.602	−1.608	−1.607	−1.605	−1.594
	2	−1.576	−1.577	−1.579	−1.582	−1.574	−1.565
	5	−1.506	−1.501	−1.502	−1.513	−1.508	−1.494
	10	−1.382	−1.375	−1.377	−1.405	−1.381	−1.365
Utilization efficiency (%)	0.5	42.62	44.21	48.72	53.15	42.37	32.66
	2	44.27	45.48	50.12	57.31	43.96	34.15
	5	48.64	47.83	51.69	59.46	48.71	36.85
	10	50.22	49.34	53.78	61.86	49.92	39.14
Specific capacity (mA h g^−1^)	0.5	867	916	955	1007	849	671
	2	1027	1133	1184	1218	994	749
	5	1143	1212	1263	1352	1102	811
	10	1218	1308	1348	1437	1196	872
Specific Energy (mW h g^−1^)	0.5	1391	1467	1536	1618	1363	1069
	2	1618	1787	1869	1927	1565	1172
	5	1721	1819	1897	2046	1662	1597
	10	1683	1798	1856	2019	1652	1190

**Table 5 materials-17-04179-t005:** Compared with the discharge performance of commercial AZ31 and AM10 magnesium alloys.

	Mg-Zn-1Sr	AZ31	AM10
Discharge potential (V)	−1.513	−1.31	−1.354
Utilization efficiency (%)	59.46	41.66	43.6
Specific capacity (mA h g^−1^)	1352	1243	1326
Specific Energy (mW h g^−1^)	2046	1628	1795

## Data Availability

The original contributions presented in the study are included in the article, further inquiries can be directed to the corresponding author.

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
