# Peer review of "Discharge Properties and Electrochemical Behaviors of Mg-Zn-xSr Magnesium Anodes for Mg–Air Batteries"

_materials, 2024, doi:10.3390/ma17174179_

Round 1

Reviewer 1 Report

Comments and Suggestions for Authors

some issues need to be solved before publication, described as follows:

+ A flow chart can be useful.

+ Please change formulas by Equations.

+ In Figure 3, scale bars are not shown correctly.

+ How the grain size was measured. Any equation or technique?

+ Please an analysis in depth is required in Figure 5. Why are the main assumptions for the differences between lines green and yellow? Also, what happened in the first two hours?. What about the fluctuations... please associate with the sem micrographs.

There is nothing more to say; good job!

+

Comments on the Quality of English Language

minor issues

Reviewer 2 Report

Comments and Suggestions for Authors

In this paper, the effect of Sr microalloying on the electrochemical and discharge properties of Mg-Zn-xSr model alloys as anodes in Mg-air batteries was investigated (in terms of a theoretical study), focusing especially on the manuscript deals with detailed insights into electrochemical behaviour, it focuses mostly on and related to influence resulted from introducing (adding) Sr as a minor element significance. The manuscript is a solid contribution to the field of energy storage materials, but there are some parts where it can be improved such that its significance and implications for future research, as well as practical applications, become clearer. Although the results are intriguing, this manuscript could be improved by a more in-depth discussion of the mechanism for how Sr modification modifies microstructure and relates to electrochemical performance. Furthermore, the discussion should be extended to a more in-depth comparison with other technologies and as well as an attitude limited by Sr's use on those alloys. The conclusion is brief, and the implications could be expanded slightly to communicate more clearly the wider significance of their results—particularly in terms of directions for future research. Below authors can find a detailed section-by section report. I strongly suggest the authors to answer to all the questions raised by the reviewer and insert all the answers properly in the final manuscript.

Abstract

The abstract of this manuscript effectively suggests the main findings and significance of the study, however, while the abstract is informative, it could benefit from increased specificity and clarity in both the methodology and the explanation of the results. The inclusion of quantitative data and a more detailed discussion of the underlying mechanisms would strengthen the abstract, making it more compelling and informative for readers. If there is no space in the abstract to properly answer the following questions, insert your answers within the body of the manuscript in the rest of the sections.

A1) Which specific electrochemical techniques were employed to assess the discharge properties of the Mg-Zn-xSr alloys, and why were these techniques chosen?

A2) What are the detailed mechanisms through which Sr addition refines the grain size and alters the passivation film composition, leading to improved electrochemical performance?

A3) Can the authors provide specific quantitative results on how the addition of Sr impacts the anode efficiency and energy density at different concentrations?

A4) How does the performance of the Mg-Zn-1Sr alloy compared to other commercially available anode materials for Mg-air batteries?

A5) What are the potential limitations or drawbacks of Sr addition, particularly at higher concentrations, and how might these affect the long-term performance of Mg-air batteries?

1. Introduction

The introduction section effectively establishes the significance of the research within the broader context of energy conservation and the development of clean energy technologies. It provides a well-structured overview of the challenges associated with Mg-air batteries and the potential role of alloying, particularly with Sr, in overcoming these challenges. However, the section would benefit from a more detailed exploration of the underlying mechanisms by which Sr addition improves the properties of Mg-Zn alloys. Additionally, incorporating quantitative data and expanding the discussion to include a comparison with other battery technologies would provide a more comprehensive foundation for the study.

1.1) What are the specific atomic or crystallographic mechanisms by which Sr addition refines the grain size and improves corrosion resistance in Mg-Zn alloys?

1.2) How does the performance of Mg-Zn-Sr alloys compare quantitatively to other Mg alloy systems (e.g., Mg-Al or Mg-Li) in the context of Mg-air batteries?

1.3) What are the specific electrochemical challenges faced by current Mg-air batteries, and how does the introduction of Sr address these challenges?

1.4) Are there any potential drawbacks or limitations associated with Sr addition in terms of mechanical properties, manufacturability, or long-term stability of the alloys?

1.5) How does the anticipated improvement in Mg-air battery performance with Sr addition compared to advancements in other battery technologies, such as lithium-ion or sodium-air batteries?

1.6) Some recent technology developments are missing and should be taken into consideration in this study such as metastructures [Inverse-designed metastructures that solve equations, Science 363 (6433), 1333-1338, 2019] and nanoparticles [Targeted dielectric coating of silver nanoparticles with silica to manipulate optical properties for metasurface applications, Materials Chemistry and Physics, 126250, 2022].

2. Materials and methods

This section is very well-described and methodologically detailed that neatly describes exactly how these experiments were done. The authors have shown great care in documenting the preparation, characterization, and testing of Mg-Zn-xSr alloys, which allows this work to be reproduced by other fellow researchers. However, it could be improved by giving more detailed explanations of the reasons behind making specific changes to experimental design and how data was processed then made reproducible.

2.1) What specific roles do the protective gases (Ar and SF6) play during the alloy casting process, and how do they influence the final microstructure of the Mg-Zn-xSr alloys?

2.2) Can you provide more detailed information on the criteria used for selecting the areas for grain size measurement in the ImageJ analysis, and how was measurement accuracy ensured?

2.3) Why were the specific temperatures (e.g., 750 °C for melting, 720 °C for homogenization) chosen during the alloy preparation, and how do these temperatures affect the microstructure and properties of the alloys?

2.4) How does the electrochemical performance of the Mg-Zn-xSr alloys compare to pure Mg or Mg-Zn alloys without Sr, and what specific improvements are observed due to Sr addition?

2.5) What statistical methods were used to analyse the reproducibility of the electrochemical and battery performance data, and how were potential outliers or anomalies handled?

3. Results and discussion

The section is well-organized and effectively presents the findings of the study, linking the microstructural, electrochemical, and discharge properties of the Mg-Zn-xSr alloys. The authors have done a commendable job of integrating various characterization techniques to build a comprehensive understanding of how Sr addition influences the performance of these alloys in Mg-air batteries. However, the section could benefit from more in-depth quantitative analysis, enhanced mechanistic discussions, and a broader comparison with existing literature. Additionally, addressing potential limitations and providing more detailed visual data representations would further strengthen the impact of this section.

3.1 Microstructures of the Mg-Zn-xSr Alloys:

This paragraph provides a thorough analysis of the microstructural evolution with increasing Sr content. The use of XRD, OM, and SEM to elucidate the effects of Sr on grain size and phase distribution is well-executed. However, more quantitative analysis and a deeper mechanistic discussion would enhance the understanding of how these microstructural changes influence the electrochemical properties.

3.1.1) How does the grain size distribution quantitatively correlate with the Sr content, and what is the statistical significance of this correlation?

3.1.2) What are the crystallographic orientations of the grains, and how might these influence the electrochemical behavior of the alloys?

3.1.3) Can the authors provide more detailed phase quantification from the XRD data, particularly in terms of phase fractions as a function of Sr content?

3.1.4) What role do grain boundary characteristics play in the corrosion resistance of these alloys, particularly at different Sr concentrations?

3.1.5) How do the observed second phases, such as Mg17Sr2 and Mg11Zn4Sr3, influence the mechanical properties of the alloys, and how might this affect their performance in Mg-air batteries?

3.2 Electrochemical Analysis of Mg-Zn-xSr Alloys:

The electrochemical analysis is detailed and well-correlated with the microstructural findings. The use of OCP, polarization curves, and EIS data provides a comprehensive picture of the electrochemical behavior of the Mg-Zn-xSr alloys. However, the section would benefit from a more detailed discussion of the mechanisms behind the observed electrochemical trends, particularly in relation to the microstructure and surface chemistry. Additionally, a comparative analysis with other common Mg-based alloys or commercial anode materials could provide a broader context for the results.

3.2.1) How does the charge transfer resistance (Rct) quantitatively relate to the microstructural features observed in the alloys, and what is the underlying mechanism?

3.2.2) What is the impact of the surface oxide film's composition and thickness on the observed OCP and polarization behaviour?

3.2.3) How do the electrochemical properties of the Mg-Zn-xSr alloys compare to those of pure Mg or other Mg-based alloys commonly used in batteries?

3.2.4) What is the influence of electrolyte composition on the electrochemical behaviour, and how might different electrolytes affect the performance of these alloys?

3.2.5) Can the authors provide more insight into the EIS data fitting process, particularly regarding the choice of equivalent circuit models and their physical significance?

3.3 Discharge Properties of the Mg-Zn-xSr Alloys:

This sub-section effectively presents the discharge performance of the alloys in Mg-air batteries, with clear correlations drawn between the microstructural characteristics and the discharge behaviour. The use of quantitative data such as specific capacity and energy density is appropriate and well-presented. However, the section could be improved by discussing the long-term stability of these alloys under real-world conditions and by comparing the results with existing battery technologies. Furthermore, exploring the impact of varying environmental conditions on discharge performance would add robustness to the conclusions.

3.3.1) How do the discharge properties of the Mg-Zn-xSr alloys compare to those of other anode materials used in commercial Mg-air batteries?

3.3.2) What is the relationship between the microstructural features (e.g., grain size, phase distribution) and the observed discharge potential stability over time?

3.3.3) How does the anode utilization efficiency change with varying Sr content under different environmental conditions, such as temperature or electrolyte concentration?

3.3.4) What are the long-term discharge behaviours of these alloys, and how does Sr addition impact their stability over extended cycling?

3.3.5) How do the observed discharge properties correlate with the specific phase content and distribution in the alloys, particularly in relation to the second phases identified in the microstructural analysis?

3.4 Surface Morphologies after Discharge:

SEM images post-discharge reveal that the results of Sr addition are clearly observable and detailed when it comes to corrosion properties related to surface morphologies. In summary, the sub-section contains an efficient connection of these morphological changes to general performance consequences within their respective anodes. However, the section could be improved by presenting detailed chemical analyses of corrosion products with techniques like EDS or XPS. Finally, a more complete picture of the performance and overall mechanical integrity as well as long-term durability of these anodes could be drawn if further investigations are conducted that would involve discussion about how surface features might affect them.

3.4.1) How does the observed surface morphology after discharge correlate with the micro galvanic effects between different phases, and what is the quantitative impact on performance?

3.4.2) What is the role of hydrogen evolution during the discharge process, and how does Sr addition influence this phenomenon?

3.4.3) How does the thickness and composition of the discharge product layer vary with Sr content, and what are the implications for long-term corrosion resistance?

3.4.4) Can the authors provide more detailed SEM-EDS analysis of the surface corrosion products to better understand the chemical processes occurring during discharge?

3.4.5) How do the observed surface morphologies influence the overall mechanical integrity of the anodes, particularly in terms of their resistance to cracking or spalling?

4. Conclusion

This final section effectively summarizes the key findings of the study, highlighting the benefits of Sr addition to Mg-Zn alloys in improving their performance as anodes in Mg-air batteries. The authors successfully communicate the significance of the Mg-Zn-1Sr alloy's performance and its potential as a high-efficiency anode material. However, the section could be further strengthened by expanding on the practical implications of the findings, outlining clear future research directions, and addressing potential limitations.

4.1) What are the specific challenges associated with scaling up the production of Mg-Zn-1Sr alloys for commercial Mg-air batteries, and how might these be addressed?

4.2) How does the performance of the Mg-Zn-1Sr alloy compared to other anode materials in real-world Mg-air battery applications, particularly in terms of long-term stability and cost-effectiveness?

4.3) What are the potential trade-offs or limitations of using Sr as an alloying element in Mg-air batteries, and how can these be mitigated in future research?

4.4) What additional alloying elements could be explored in combination with Sr to further enhance the performance of Mg-Zn alloys, and what mechanisms might these elements influence?

4.5) How do the findings of this study contribute to the broader development of sustainable energy storage technologies, particularly in the context of reducing reliance on rare or expensive materials?

Round 2

Reviewer 2 Report

Comments and Suggestions for Authors

The authors answered clearly the reviewer’s concerns.

New interesting applications and future works can be envisioned.